# Validity and reliability of the XSENSOR in-shoe pressure measurement system

**Daniel Parker** [ID]*, **Jennifer Andrews, Carina Price** [ID]

School of Health and Society, University of Salford, Salford, United Kingdom

* d.j.parker1@salford.ac.uk

## Abstract

### Background

In-shoe pressure measurement systems are used in research and clinical practice to quantify areas and levels of pressure underfoot whilst shod. Their validity and reliability across different pressures, durations of load and contact areas determine their appropriateness to address different research questions or clinical assessments. XSENSOR is a relatively new pressure measurement device and warrants assessment.

### Research question

Does the XSENSOR in-shoe pressure measurement device have sufficient validity and reliability for clinical assessments in diabetes?

### Methods

Two XSENSOR insoles were examined across two days with two lab-based protocols to assess regional and whole insole loading. The whole insole protocol applied 50–600 kPa of pressure across the insole surface for 30 seconds and measured at 0, 2, 10 and 30 seconds. The regional protocol used two (3.14 and 15.9 $cm^2$ surface area) cylinders to apply pressures of 50, 110 and 200 kPa to each insole. Three trials of all conditions were averaged. The validity (% difference and Root Mean Square Error: RMSE) and repeatability (Bland Altman, Intra-Class Correlation Coefficient: ICC) of the target pressures (whole insole) and contact area (regional) were outcome variables.

### Results

Regional results demonstrated mean contact area errors of less than 1 $cm^2$ for both insoles and high repeatability ($\geq$0.939). Whole insole measurement error was higher at higher pressures but resulted in average peak and mean pressures error < 10%. Reliability error was 3–10% for peak pressure, within the 15% defined as an analytical goal.

### Significance

Errors associated with the quantification of pressure are low enough that they are unlikely to influence the assessments of interventions or screening of the at-risk-foot considering

**Data Availability Statement:** All relevant data are within the manuscript and its Supporting information files.

**Funding:** The author(s) received no specific funding for this work.

**Competing interests:** The authors have declared that no competing interests exist.

clinically relevant thresholds. Contact area is accurate due to a high spatial resolution and the repeatability of the XSENSOR system likely makes it appropriate for clinical applications that require multiple assessments.

## Introduction

In-shoe pressure measurement systems are used in clinic and research to quantify contact area and pressure at the foot-shoe and foot-insole interface [1]. Plantar pressure data may be used to quantify in-shoe pressures during specific tasks [2], in specific populations [3], or to optimise interventions such as therapeutic footwear for diabetic populations [4]. More recently clinical practice guidelines have taken the step to recommend the use of therapeutic footwear with a demonstrated plantar pressure relieving effect [5], reinforcing the need for effective, valid and repeatable (within- and between-day) methods of measurement.

Several in-shoe pressure measurement systems are commercially available. Their suitability to specific research or clinical use is determined by sensor type, size, resolution, and distribution and how these align with the data-collection protocol and intended application. Capacitive sensors, for example, succumb to creep when exposed to extended periods of loading, which may require correction in a static or fatiguing protocol [6]. This would make them less suitable as devices to assess static balance tasks, however these may be perfectly suitable for use within a study that exclusively observes dynamic tasks. Devices may also be selected for their ability to identify small changes in pressure at particularly high- or low-pressure ranges, for example, when modifying ulceration risk within an at-risk foot or observing contact area changes in response to orthotic or footwear intervention [4, 7]. Assessments of intervention effectiveness typically focus on analysis of load distribution and peak or mean pressure reduction, across discrete foot regions, averaged over multiple mid-gait steps [8]. Contact area, calculated as the number of active sensors allows evaluation of changes to load distribution driven by the geometry of the intervention [8]. Peak plantar pressure, normally calculated as the absolute highest pressure for a given region during stance allows for assessment of exposure to potential risk [9]. While mean plantar pressure, calculated as the average pressure across sensors within a discrete region during stance, incorporates changes to contact area and magnitude to evaluate the broader effect on load distribution [10].

Owing to the requirement for assessment of intervention effectiveness, it is crucial that systems are demonstrated to perform in a repeatable way. The validity and reliability of some in-shoe measurement devices have been investigated utilising both bench-top [11, 12] and in-situ methods [13] through protocols with varying methodologies. High repeatability with the Pedar in-shoe system has been demonstrated between days [13, 14] and the measurement of midfoot pressure and contact area variables also demonstrate moderate to high intra class correlation coefficients (ICC = 0.556–0.947 and 0.529–0.921 respectively) between trials [15]. The Tekscan system has been reported to have low durability and to demonstrate significant creep (19% within 15 min) and hysteresis, high variability between and within sensors and low overall repeatability [16]. A more recent publication with their updated F-Scan in-shoe system however demonstrated reliability values that were higher, with contact area in particular having high ICCs (0.91–0.98) and the forefoot and toe region also being highly repeatable values for pressure time integral and average and peak pressure (ICC = 0.83–0.98) [17]. This was consistent with findings with the same system in a diabetic cohort where peak pressure coefficient of variations (CV 0.150, 0.155) and ICCs (0.755 and 0.751) of the metatarsal heads had the best

indices of reliability [18]. Most more recent literature demonstrates systems produce high correlations or strong relationships between two days or measures.

These studies highlight that consideration of appropriate technical specification of the in-shoe pressure system is required prior to selecting a system for use in clinic and for research purposes. To evaluate a plantar pressure system for clinical use and benchmark performance it is important to consider the expected range, the observed change with intervention and the minimally important difference (MID) across repeated measures [19, 20]. MID and observed change in plantar pressure measurement is most clearly established in the context of offloading the diabetic foot and international guidelines for prevention of foot ulcers provides a strong recommendation for reductions of 30% peak plantar pressure for effective intervention based on moderate quality of evidence from the literature [5].

In 2016, Price et al., published an assessment of the validity and repeatability of three widely used in-shoe pressure systems (Medilogic (T&T Medilogic, Medizintechnik Gmb Schönefeld, Germany), Pedar (Novel, Munich, Germany) and Tekscan (Tekscan Inc., Boston, USA)) for a range of pressure magnitudes and durations [21]. Combining this prior protocol for objective pressure system evaluation with the known requirements for 30% reduction to detect clinically meaningful differences for intervention in diabetic patients facilitates a benchmark for appropriateness for clinical assessments. The aim of this study is to assess the validity and between-day reliability of the XSENSOR X4 system (XSENSOR Technology Corporation, Calgary, Canada) with the aim of determining its suitability for clinical assessments in diabetes.

## Method

The XSENSOR X4 Insole System is a wireless in-shoe pressure measurement system, available in UK sizes 4–15 (Fig 1). Insoles have a standard sampling rate of 75 Hz and 230 capacitive sensels (resulting in a resolution: 7–9 mm), reportedly recording pressures from 7–883 kPa with ±5% accuracy of the full-scale calibration. Two XSENSOR insoles (UK 4/5 and 9/10),

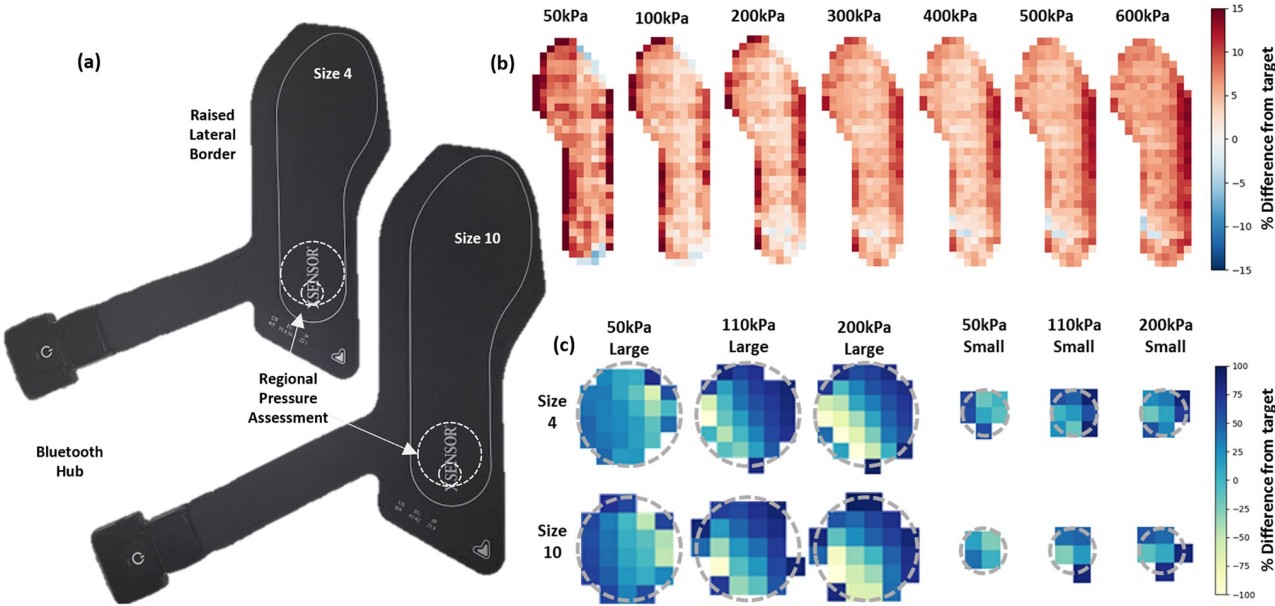

**Fig 1.** (a) XSENSOR insole showing location of placement of cylinders for regional assessment (dashed lines). (b) Average peak pressure maps for difference from target pressures for whole insole protocol. (c) Regional protocol, with cylinder area overlay (dashed lines) sensor size has been scaled to correct for differences between XSENSOR insole sizes.

hereafter referred to as S4 and S10, underwent the same protocol as previously described to assess three other commercially available insole systems [21]. As is standard with capacitive systems, sensel number is consistent across insoles and therefore sensel area of the XSENSOR insoles varied between the two insole sizes (S4 = 0.49 cm$^2$ and S10 = 0.67 cm$^2$). Insoles were supplied with a calibration file for use in XSENSOR software. A whole insole and a regional protocol were applied to the sensors and throughout both protocols, sensels which registered >10 kPa were deemed active sensels and included in analysis.

## Whole insole protocol

The TruBlue calibration device (Novel, Munich, Germany) was used to load each insole evenly across its surface at a range of pressures (50, 100, 200, 300, 400, 500 and 600 kPa). Each pressure was applied manually as quickly as possible using a hand release valve to inflate the internal bladder. The start of the measurement was when the target pressure was achieved (monitored by a pressure gauge; VDO Instruments, Germany).

Each pressure was applied for 30 seconds (maintained at ±2% target pressure, monitored by a pressure gauge). In each of the seven target pressure conditions three measurements were recorded, and an average mean and average peak pressure were calculated across the insole. Based on manufacturer guidance some sensels on the periphery of the insole were excluded from analysis due to a stepped border enclosing sensel cabling which was expected to affect the local application of even pressure.

## Regional protocol

Two cylinders were used to apply regional loads in this study. Each cylinder was made of rigid plastic and had a 1mm layer of PORON at its base. Cylinders were applied to the rearfoot of each insole, where we would anticipate the heel to be making contact (Fig 1a). Cylinder areas were chosen to represent anatomical sites on the foot, specifically the calcaneus (surface area 15.9 cm$^2$) and the metatarsal head (surface area 3.1 cm$^2$). Cylinders were loaded by the application of weights to their top surface and supported by a vertical linear bearing to ensure load through their centre to generate realistic pressures for these anatomical regions during stance of 50, 110 and 200 kPa. For each of the six conditions (2 areas x 3 pressures) three measurements were recorded of 30 seconds.

## Variables

Variables were calculated for the regional and whole insoles protocol using custom-written scripts in Python (3.7.3, Numpy 1.18.1, Pandas 0.25.3).

Within the regional protocol average contact area was calculated as the cumulative area of active sensels at T0, T2, T10 and T30 of the three 30 second trials. T0 reflected a measure at the first instant at which the required pressure had been applied and the other T values added to this. Validity was established by calculation of root mean square error (RMSE) when compared to the known surface area of each cylinder (3.1 and 15.9 cm$^2$), defined as area error (AErr). Time dependence was assessed by calculating the change in area over time. Repeatability between day was calculated using intra-class correlation coefficients (ICC; two-way mixed absolute agreement) for values recorded at 2 seconds.

Within the whole insole protocol the repeatability and validity of the held load at 0, 2, 10 and 30 seconds were outcome variables (T0, T2, T10 and T30). Peak and mean pressure measurement errors (MErr) were calculated based on the absolute value versus the target pressure, also presented as a percentage of the target pressure. Validity was established by calculation of RMSE across all active sensels to the target pressures applied in the TruBlue device.

Consistency of the measured pressure across the insole was assessed by calculating the percentage of sensors within 5% of the all sensel mean. Time dependence was assessed by calculating the change in peak and mean pressure over time. Reliability was explored for the data by comparing day one to day two for each pressure (between day) and both days to the target value. These were then considered for each pressure separately then a combined dataset across all target pressures. Correlation was used to identify agreement, paired t-test to assess statistically significant bias and Bland-Altman plots with 95% limits of agreement to display combined systematic bias and random error between tests [20], defined as Reliability error (RErr). To assess limits of agreement the absolute difference and mean were calculated for each measure, difference data was assessed for normality (Shapiro-Wilk) and heteroscedastic error (correlation coefficient of absolute difference vs mean) as per the method of Nevill and Atkinson (1997) [22]. Due to the presence of heteroscedastic error in all cases, natural logarithms were taken for measurement data, differences between log transformed measures were then confirmed as normally distributed. Systematic bias was calculated as the average difference between days and an antilog was taken to calculate the mean bias ratio. Limits of agreement were calculated as agreement ratios between which 95% of difference between days can be calculated [22]. Using the 30% reduction guideline [5] we have defined an analytical goal of 15% reliability error (RErr) (combined systematic bias and random error) such that each days measured value should not be affected by more than half of the objective reduction. All statistical analysis was performed in Excel (V2108, Microsoft, USA) and Statistical Software for the Social Sciences (SPSS; V20, IBM, USA).

## Results

### Regional protocol

Measured contact area had absolute AErr mean ± S.D. of 0.86 ± 0.41 cm$^2$ for S4 and 0.68 ± 0.57 cm$^2$ for S10 across both cylinder sizes (Fig 2). When assessed based on cylinder size, the large cylinder absolute AErr was 0.91 ± 0.56 cm$^2$ while the small cylinder was 0.63 ± 0.40 cm$^2$. Over the 30s duration contact area showed either no change or a minimal increase (0 to 2%) with the exception of one measure day 1, S4, Small cylinder at 50 kPa which exhibited a 6% reduction in measured area. The contact area between-day repeatability demonstrated high values for S4 (ICC = 0.972) and S10 insoles (ICC = 0.939).

### Whole insole protocol

Peak pressure was higher than the target pressure in S4 (Absolute MErr Mean ± SD and % Error across all pressure levels: 20.64 ± 13.69 kPa; 7%) and S10 (23.09 ± 15.92 kPa; 9%). Percentage MErr were highest at 50 kPa (13% and 20% respectively) with this latter value in the S10 insole on day two being the only MErr which exceeded our analytical goal of 15% (Fig 3; Table 1). Peak pressure error increased slightly in all trials (0.6 to 2.8%) over the 30s. Mean pressure MErr was negative for S4 (7.42 ± 7.97 kPa; -2%) and S10 (15.21 ± 11.01 kPa; -4%) (Fig 3). Mean pressure MErr as a percentage was largest at 600 kPa pressure (-4% and -6%), but no values exceed our analytical goal (Fig 3; Table 2). Over the 30s duration mean pressures increased slightly in all trials (0.4 to 3%). RMSE across all sensels was lowest in S4 (mean 11.99 kPa, max 27.74 kPa) compared to S10 (mean 18.35 kPa, max 40.46 kPa) and increased in both insoles with target pressure (Fig 3).

Pressure data from day one and day two is reported for peak (Fig 4; Table 1) and mean (Table 2) pressures alongside reliability measures for the between days comparison assessed at each magnitude of applied pressure (Data is reported in full in S1 File). In all cases a positive systematic bias was observed for measurements between days along with a significant

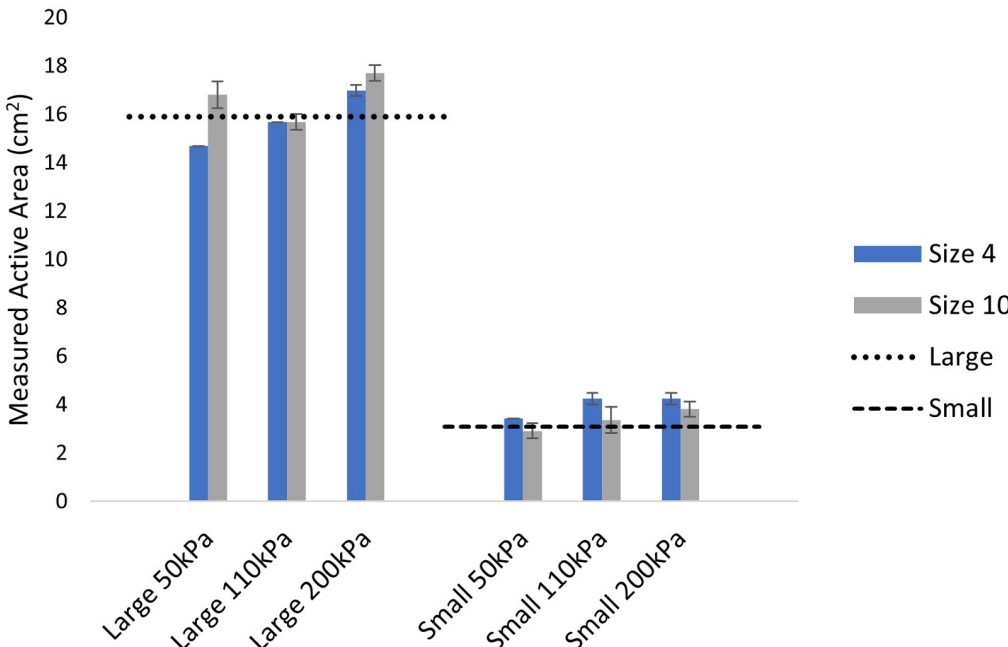

**Fig 2. Contact area for each insole and cylinder size for day two across the 2s trial.** Error bars denote standard deviation across three trials.

difference between days at many target pressure levels, the bias ratio ranged from 1.02 to 1.06 in peak pressure and 1.02 to 1.07 in mean pressure data. Considering the reliability of the data across all target pressures, for S5 (correlation post log transform of measurement error and mean -0.57, t-test of day 1 and 2 p < .001) and s10 (correlation post log transform of measurement error and mean -0.65, t-test of day 1 and 2 p < .001).

Applying the 95% ratio limits of agreement to evaluate against the MID, for a peak pressure assessed at a 200 kPa level on day 1, it is possible that the XSENSOR system could obtain an estimate as low as 200x1.01 = 202 kPa or as high as 200x1.10 = 220 kPa on day 2 (S4, Table 1). This represents a maximum RErr of 20 kPa, or 10%, which did not exceed our analytical goal and was considerably lower than the MID of 60 kPa or 30% at 200 kPa, for S10 the maximum RErr was lower at 8%. Assessment of the worst-case scenario within mean pressure measures can be made at the 50 kPa level (Table 2). Assuming 50 kPa on day 1 the lower and upper bounds of reassessment would be 50x0.96 = 48 kPa and 50x1.15 = 57.5 kPa on day 2, the maximum RErr was equal to our analytical goal at this level.

The sensors across the insole demonstrated high consistency, particularly for the S4 insole (Table 3). At pressures above 50 kPa the number of sensels within 5% of the mean pressure value was >80% for both insole sizes (Table 3).

## Discussion

Peak pressure is routinely used within clinical evaluation to establish high-risk regions, however is sensitive to error from single sensels within a discrete region [23]. The grand average measurement error across target loads and days for mean pressure was 4% in S4 and 6% in S10 and for peak pressure was 5% in S4 and 7% in S10. This grand average using the XSENSOR system was comparable to grand average measurement errors reported for Pedar (mean 4.5% and peak 4.8%) and lower than for Medilogic (mean 10.7% and peak 46.2%) and Tekscan

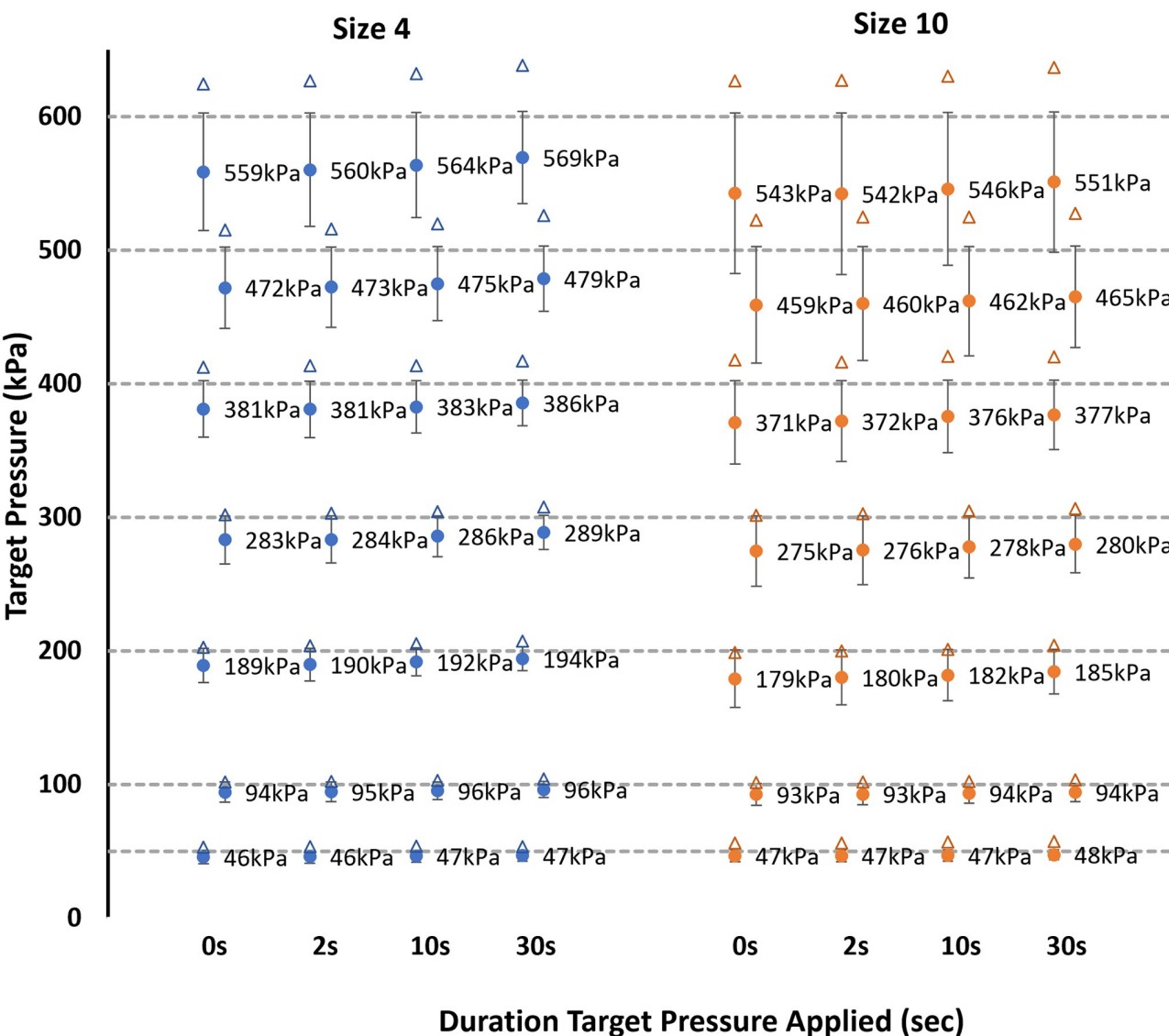

**Fig 3. Peak and mean pressure values recorded over 0, 2, 10 and 30 s for the 7 target pressures for both insoles on day one.** Where circles mark the mean pressure data, error bars denote the Root Mean Square Error (RMSE) across all individual sensels about the mean recorded value and the triangle marks the peak pressure data.

(mean 60.5% and peak 193.3%) using the same protocol [21]. Contrasting the results for other insole systems in the prior study [21], mean pressure error increased with increasing target pressure, with very good agreement (<5% error) at low pressures (50–200 kPa) which could improve accuracy of contact area variables. The RMSE mean (12–18 kPa) and max (28–40 kPa) was higher than reported for Pedar (2.5, 4.7 kPa) but comparable to Medilogic (mean 28.5, max 45.7 kPa) and Tekscan (25.5, 41.8 kPa).

Within this study the analysis has extended to assessment with a clinically relevant objective. Within a clinically relevant range (100–400 kPa) [9] XSENSOR insoles recorded peak pressure measurement errors of 1 to 8% and mean pressure measurement errors in the range -10 to 0%. Systematic bias between days was within 2 to 6% for peak and mean pressure while Reliability error (including random error between days) increases this to an upper limit

**Table 1. Average peak pressure at each target pressure at 2 seconds for Day 1 and Day 2 the differences, and the "ratio limits of agreement" alongside the t-test between the measures of day 1 and 2.**

| Peak Pressure | Measured Values | | | | Log Transformed Calculations | | | |
| --- | --- | --- | --- | --- | --- | --- | --- | --- |
| | Day 1 | | Day 2 | | T-Test Day1 Day 2 | Ratio Limits | | |
| | Mean±SD | %MErr | Mean±SD | %MErr | | Bias limits | Upper | Lower |
| Size 4 (S4) | | | | | | | | |
| 50 kPa | 53.59±1.04 | 7% | 56.31±0.98 | 13% | 0.01 | 1.05*/÷1.02 | 1.07 | 1.04 |
| 100 kPa | 102.44±5.20 | 2% | 106.27±2.27 | 6% | 0.33 | 1.04*/÷1.08 | 1.13 | 0.96 |
| 200 kPa | 204.05±3.83 | 2% | 215.11±4.42 | 8% | 0.09 | 1.05*/÷1.05 | 1.10 | 1.01 |
| 300 kPa | 303.19±3.88 | 1% | 318.30±4.57 | 6% | 0.01 | 1.05*/÷1.01 | 1.06 | 1.04 |
| 400 kPa | 413.75±2.78 | 3% | 421.35±1.67 | 5% | 0.04 | 1.02*/÷1.01 | 1.03 | 1.01 |
| 500 kPa | 515.90±6.69 | 3% | 528.05±5.40 | 6% | 0.23 | 1.02*/÷1.04 | 1.06 | 0.99 |
| 600 kPa | 626.71±6.91 | 4% | 649.09±2.80 | 8% | 0.03 | 1.04*/÷1.02 | 1.05 | 1.02 |
| Size 10 (S10) | | | | | | | | |
| 50 kPa | 56.55±2.45 | 13% | 60.13±0.18 | 20% | 0.20 | 1.06*/÷1.10 | 1.17 | 0.97 |
| 100 kPa | 102.10±1.20 | 2% | 105.57±1.94 | 6% | 0.02 | 1.03*/÷1.01 | 1.05 | 1.02 |
| 200 kPa | 200.02±2.61 | 0% | 211.52±1.85 | 6% | 0.02 | 1.06*/÷1.02 | 1.08 | 1.03 |
| 300 kPa | 302.83±6.07 | 1% | 315.33±3.43 | 5% | 0.03 | 1.04*/÷1.02 | 1.06 | 1.02 |
| 400 kPa | 416.53±5.77 | 4% | 428.24±2.42 | 7% | 0.06 | 1.03*/÷1.02 | 1.05 | 1.01 |
| 500 kPa | 524.70±1.33 | 5% | 539.00±6.54 | 8% | 0.12 | 1.03*/÷1.03 | 1.06 | 1.00 |
| 600 kPa | 627.11±8.82 | 5% | 651.83±2.56 | 9% | 0.03 | 1.04*/÷1.02 | 1.06 | 1.02 |

Measured values were transformed using natural logarithms to obtain ratio limits of agreement. %MErr: Measurement error, percentage error from the target pressure, Normality: Significance value of Shapiro-Wilk, T-Test: two tailed, paired samples

**Table 2. Average mean pressure at each target pressure at 2 seconds for Day 1 and Day 2, the differences, and the "ratio limits of agreement" alongside the t-test between the measures of day 1 and 2.**

| Mean Pressure | Measured Values | | | | Log Transformed Calculations | | | |
| --- | --- | --- | --- | --- | --- | --- | --- | --- |
| | Day 1 | | Day 2 | | T-Test Day1 Day 2 | Ratio Limits | | |
| | Mean±SD | %MErr | Mean±SD | %MErr | | Bias limits | Upper | Lower |
| Size 4 (S4) | | | | | | | | |
| 50 kPa | 46.13±0.95 | -8% | 49.36±0.90 | -1% | 0.02 | 1.07*/÷1.02 | 1.10 | 1.04 |
| 100 kPa | 94.70±4.72 | -5% | 98.25±1.40 | -2% | 0.34 | 1.04*/÷1.09 | 1.13 | 0.96 |
| 200 kPa | 189.91±4.06 | -5% | 199.98±3.54 | 0% | 0.10 | 1.05*/÷1.05 | 1.11 | 1.00 |
| 300 kPa | 283.61±3.25 | -5% | 297.84±4.17 | -1% | 0.01 | 1.05*/÷1.01 | 1.06 | 1.04 |
| 400 kPa | 380.95±2.14 | -5% | 389.60±1.84 | -3% | 0.01 | 1.02*/÷1.01 | 1.03 | 1.02 |
| 500 kPa | 472.52±5.62 | -5% | 485.38±3.73 | -3% | 0.16 | 1.03*/÷1.03 | 1.06 | 0.99 |
| 600 kPa | 560.25±6.93 | -7% | 577.64±2.16 | -4% | 0.06 | 1.03*/÷1.02 | 1.05 | 1.01 |
| Size 10 (S10) | | | | | | | | |
| 50 kPa | 46.89±2.07 | -6% | 49.16±0.14 | -2% | 0.28 | 1.05*/÷1.10 | 1.15 | 0.96 |
| 100 kPa | 92.99±0.83 | -7% | 94.90±1.17 | -5% | 0.02 | 1.02*/÷1.01 | 1.03 | 1.01 |
| 200 kPa | 180.42±2.03 | -10% | 190.67±2.85 | -5% | 0.03 | 1.06*/÷1.03 | 1.09 | 1.03 |
| 300 kPa | 275.56±6.81 | -8% | 287.13±3.20 | -4% | 0.05 | 1.04*/÷1.03 | 1.07 | 1.01 |
| 400 kPa | 372.30±4.64 | -7% | 381.46±2.21 | -5% | 0.06 | 1.02*/÷1.02 | 1.04 | 1.01 |
| 500 kPa | 460.20±1.94 | -8% | 475.44±6.42 | -5% | 0.12 | 1.03*/÷1.03 | 1.07 | 1.00 |
| 600 kPa | 542.41±6.99 | -10% | 564.77±3.20 | -6% | 0.02 | 1.04*/÷1.01 | 1.06 | 1.03 |

Measured values were transformed using natural logarithms to obtain ratio limits of agreement. %MErr: Measurement error, percentage error from the target pressure, Normality: Significance value of Shapiro-Wilk, T-Test: two tailed, paired samples.

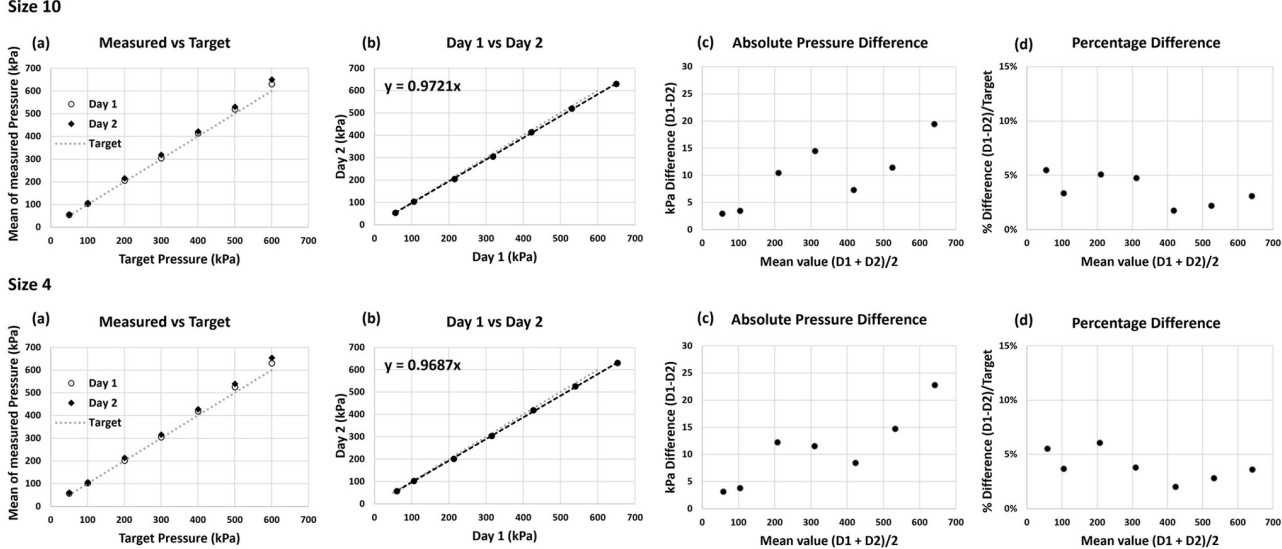

**Fig 4. Peak pressure values recorded at 2s for the 7 target pressures in both insoles between days.** (a) Proximity of measured values to the target (dashed line). (b) Relation between measured peak pressure on Day 1 and Day 2 showing slope of curve. (c) Average absolute difference (error) between days plotted against between day mean values (d) Average difference between days as percentage of target against between day mean values.

between 3 to 10% for peak pressure and 3 to 11% for mean pressure. Between day reliability showed significant differences when assessed across the whole operating range however at individual pressure levels these differences were not meaningful when assessed against the analytical goal. The overall error observed with XSENSOR is not expected to bias evaluations of efficacy, given that an effective intervention is expected to reduce pressure by more than 30% [5]. Applying this to a case from the literature; Preece et al 2017 observed a 37% reduction from an initial peak plantar pressure level of ~260 kPa by optimising apex position on rocker soled footwear [24]. Using XSENSOR in this setting, a maximum Reliability error (RErr) of 10% would not alter the interpretation of the assessment of these interventions. Mean pressure however consistently underestimated the target pressure, with a larger underestimation at higher pressures, which points to the importance of peak pressure when measuring at-risk feet [1, 23]. However, peak pressures at high pressures (>400kPa) have higher absolute errors, particularly in the S10 insole. Although pressures of this magnitude may be rare in standard shod walking, the increased variability of peak pressure measurement in this range should be

**Table 3. Percentage of sensels within 5% of mean value for all pressures and times and each insole size and day 1 and 2.**

| Target pressure | | Size 4 | | | | | | | | Size 10 | | | | | | | |
| --- | --- | --- | --- | --- | --- | --- | --- | --- | --- | --- | --- | --- | --- | --- | --- | --- | --- |
| | | Day 1 | | | | Day 2 | | | | Day 1 | | | | Day 2 | | | |
| Magnitude (kPa) | Duration (sec) | 0 | 2 | 10 | 30 | 0 | 2 | 10 | 30 | 0 | 2 | 10 | 30 | 0 | 2 | 10 | 30 |
| 50 | | 78 | 77 | 78 | 79 | 80 | 81 | 81 | 81 | 76 | 77 | 77 | 78 | 77 | 76 | 76 | 77 |
| 100 | | 86 | 86 | 87 | 86 | 88 | 88 | 87 | 87 | 84 | 84 | 84 | 84 | 84 | 84 | 85 | 85 |
| 200 | | 87 | 88 | 88 | 88 | 89 | 90 | 89 | 89 | 84 | 84 | 83 | 84 | 83 | 84 | 83 | 84 |
| 300 | | 94 | 95 | 96 | 96 | 96 | 96 | 97 | 97 | 90 | 90 | 91 | 92 | 91 | 91 | 92 | 91 |
| 400 | | 94 | 95 | 95 | 95 | 94 | 94 | 93 | 94 | 87 | 88 | 88 | 88 | 89 | 89 | 89 | 89 |
| 500 | | 93 | 94 | 93 | 94 | 92 | 92 | 93 | 92 | 88 | 88 | 88 | 88 | 89 | 89 | 88 | 89 |
| 600 | | 93 | 93 | 93 | 92 | 90 | 91 | 90 | 91 | 89 | 90 | 90 | 89 | 90 | 89 | 89 | 90 |

considered. Tasks which include more dynamic activities, higher velocities as well as those with low contact areas such as high heels produce peak pressures within these ranges [25]. This is particularly relevant with the trend to real-world data collection which is arising in more recent literature, where peak pressures may exceed those recorded in level walking [26]. The number of sensels reading within 5% of the mean pressure (Table 1), an indicator of the insole's consistency, was low at 50 kPa (75–80%), but increased to an optimal level (>90%) at 300 kPa target pressure, thereafter reducing but maintaining high numbers of sensels (>87%). The size 10 insole showed lower consistency, which likely underpins the greater variability observed in mean and peak pressure for this insole.

A good representation of applied contact area was observed with the large cylinder for all pressures, with less than 11% error across all pressures. Representation of the small cylinder showed greater surface area error (max 37%) with a tendency to overestimate contact area, suggesting the involvement of additional peripheral sensels. For context in a test with the larger cylinder addition or removal of 1 peripheral sensel would represent 3% error in S4 and 5% error in S10 and as such an 11% error could be said to represent 3–4 sensels. For the small cylinder a 1 sensel change has a much greater effect with errors of 16% in S4 and 24% in S10, representing 2–3 sensels. For comparable conditions (Large 50, Large 200 and Small 200) XSENSOR demonstrated better area estimation than prior systems assessed with the same protocol [21]. This improved accuracy is underpinned by the high spatial resolution of the XSENSOR system with a minimum of 1.5 sensors per $cm^2$ (1.5 / $cm^2$ for S10, 2.0 / $cm^2$ for S4). This represents a higher density than Medilogic (0.79 / $cm^2$) and Pedar (0.57–0.78 / $cm^2$) systems, but a reduced density when compared to Tekscan (3.9 / $cm^2$). The area error is relevant for assessment of localised regions of interest, such as prior ulceration sites, which are typically small plantar areas with high associated pressures. This could impact on both contact area calculation and the application of average pressure measurement. However, these data should be considered alongside the limitations of the protocol and the application of an isolated rigid cylinder which is not a realistic representation of foot anatomy and pressure distribution. The interaction of pressure from multiple contact points between the foot and the ground during gait may negate the activation of boundary sensels and their influence on measured contact area within this regional protocol. However, the observed involvement of these boundary sensels surrounding regions of high pressure suggests a potential to influence measured pressure in neighbouring anatomical regions. Between-day reliability of the system was high when considering regional loading (at 2s) for both cylinder and insole sizes assessed by the ICCs.

The whole insole loading (pressures compared across days) demonstrated measurement error and reliability error within our analytical goals. The regional loading demonstrate high between-day reliability and lower area errors than evident with other pressure systems. This therefore, suggests that the XSENSOR system is appropriate for repeated measures visits which aim to identify localised changes in pressure or compare between interventions across clinical visits for example. Also, that this reliability would not be substantially influenced by the insole size used. These data however, have the limitation that they were collected within controlled laboratory conditions and the reliability of the system for clinical use should be explored in-situ with relevant patients and relevant interventions also.

## Limitations

The whole insole protocol utilised a pressurised bladder to apply an even load to the insole surface, however this does not account for the structure of the XSENSOR insole which required some omission of data post-collection. The target pressure varied slightly but was always within 2% of the desired load. The effect of this on the outcomes cannot be isolated. The

contact surfaces used to produce the regional loading responses were flat and solid and as such did not fully represent the loading applied by the soft tissues of the foot or the interaction with footwear materials that would be evident in-shoe. The influence of in-shoe factors such as temperature and bending were not considered and may affect the systems differently. The reliability of the system during a gait assessment should be explored.

## Supporting information

**S1 File. Additional between day figures.** This file contains figures for peak pressure measures between days at each magnitude of applied pressure using the mean of the three trials for each time point (T0, T2, T10, T30).
(DOCX)

**S2 File. Data underlying findings.** This excel file contains the dataset used for analysis. 5 tabs are presented in including summary data tables for: 1) Day 1 data for the whole insole, 2) Day 2 data for the whole insole, 3) Calculations for peak pressure reliability, 4) Calculations for mean pressure reliability, 5) Regional analysis of contact area.
(XLSX)

## Acknowledgments

XSENSOR Technology Corporation (Calgary, Canada) provided the insole system for testing but had no involvement in the study design, collection, analysis or interpretation of data or the manuscript writing or submission.

## Author Contributions

**Conceptualization:** Daniel Parker, Carina Price.

**Data curation:** Daniel Parker.

**Formal analysis:** Daniel Parker, Jennifer Andrews, Carina Price.

**Investigation:** Daniel Parker, Jennifer Andrews, Carina Price.

**Methodology:** Daniel Parker, Jennifer Andrews, Carina Price.

**Writing – original draft:** Daniel Parker, Jennifer Andrews, Carina Price.

**Writing – review & editing:** Daniel Parker, Jennifer Andrews, Carina Price.

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
