## [Decision Letter · Decision Letter 0]

21 Jan 2022

PONE-D-21-30139Validity and repeatability of the XSENSOR in-shoe pressure measurement systemPLOS ONE

Dear Dr. Parker,

Thank you for submitting your manuscript to PLOS ONE. After careful consideration, we feel that it has merit but does not fully meet PLOS ONE’s publication criteria as it currently stands. Therefore, we invite you to submit a revised version of the manuscript that addresses the points raised during the review process.

Please attend carefully to both reviewers' comments and address particularly the concerns of reviewer 2.

We look forward to receiving your revised manuscript.

Kind regards,

Peter Andreas Federolf

Academic Editor

PLOS ONE

https://journals.plos.org/plosone/s/file?id=ba62/PLOSOne_formatting_sample_title_authors_affiliations.pdf”

2. Please note that in order to use the direct billing option the corresponding author must be affiliated with the chosen institute. Please either amend your manuscript to change the affiliation or corresponding author, or email us at plosone@plos.org with a request to remove this option.

Reviewers' comments:

Reviewer's Responses to Questions

**Comments to the Author**

1. Is the manuscript technically sound, and do the data support the conclusions?

Reviewer #1: Yes

Reviewer #2: No

2. Has the statistical analysis been performed appropriately and rigorously? 

Reviewer #1: No

Reviewer #2: I Don't Know

3. Have the authors made all data underlying the findings in their manuscript fully available?

Reviewer #1: Yes

Reviewer #2: Yes

4. Is the manuscript presented in an intelligible fashion and written in standard English?

Reviewer #1: Yes

Reviewer #2: Yes

5. Review Comments to the Author

Reviewer #1: Thank you for the opportunity to review the article titled Validity and repeatability of the XSENSOR in-shoe pressure measurement system. This manuscript describes a technical study of an in-shoe pressure measurement system to validate its measurement capabilities. The results of the study indicate that the XSENSOR system provides reasonable estimates of pressure and provide a characterization of bias and reliability.

General Comments

Overall, this is a well-designed and executed study.

The authors’ approach to quantifying accuracy is well founded and executed. However, I think the approach to day-to-day reliability is not as rigorous. I am generally skeptical of ICCs and their use in quantifying reliability as they have many issues that have been identified in literature (Muller and Buttner, 1994). My suggestion would be to reframe the problem of reliability as a hypothesis test with respect to a minimally important difference (MID).

The authors have provided a reference that suggests clinically important changes in pressure would be on the order of 30%. Using this (or other prior work as appropriate), we can define an MID that we would expect to see in a clinical study. If we wish to be conservative, we could use the lower end of the typical range of pressure values, 100 kPa, and the reference of 30% gives us an MID of 30 kPa. We can then take the measurements for day 1 and day 2, across all pressure conditions, and treat these as repeated measures observations. In doing so, we can calculate the distribution of the differences between days and conduct a single-sided hypothesis test with respect to an MID of 30 kPa (the alternate hypothesis being that our estimated mean of differences exceeds 30 kPa).

The advantage of this approach is that it provides a concrete benchmark for evaluating whether the insoles will measure pressure reliably. In my opinion, this is more informative than ICCs, and a more statistically rigorous test of insole reliability. I also suspect that the findings would favour the insoles as being highly reliable.

A table that provides comparative values for the XSENSOR system, similar to Table 1 from Price et al., 2016, might be of interest to the reader.

Specific Comments

The introduction is a bit meandering and some of the information is related, but tangential to the main purpose of the article. There are somewhat contradictory statements as well. It is stated that comparing between different studies/systems is very difficult, while the aim of the current study is to provide such a comparison between XSENSOR and other systems. Yet, this study does not address this difficulty at all; it only provides similar quantifications for the XSENSOR as has already been done for other systems in earlier work. I’d suggest a revision of the introduction to make it more focused on the purpose of the article: evaluating the XSENSOR system for clinical use.

Throughout the manuscript there are no spaces between numbers and units and these should be added throughout (e.g., 400kPa should be 400 kPa).

Line 53: capitalization of diabetic not necessary

Line 60: single quotes around creep not required; this is a standard physics term

Line 106: single quotes around calibration file not required

Line 108: single quotes around active sensels not required

Line 126: this sentence is a little unclear as it may read as though there were two separate cylinders with different bases; perhaps consider rewording

Line 130: “For [each of] the six conditions…”

Line 138 and 144: As mentioned in my general comments, ICCs are not the ideal representation of reliability

Line 144: suggest “performed” instead of “undertaken”

Line numbers stopped after page 10

Page 11, First paragraph, the peak pressures for 100-400 kPa measurements seem to be within 1-8 % of the target measurements, unless there is some averaging done between day 1 and 2.

Page 11, First paragraph, “… interact with evaluations of efficacy…” might be clearer as “… bias evaluations of efficacy…”

Page 11, First paragraph, while some of the mean measurements did indeed show less error than peaks, this was not universally true; I suggest rephrasing this statement

Page 13, Second paragraph, given that the testing in this study was done under very controlled laboratory conditions, the evaluation of reliability should reflect the caveat that calibration and measurement under in situ conditions might be less reliable than what has been shown

Figure 1, the scales for the grids in this figure is not obvious, and if I am correct, the scales are not identical between S4 and S10, even though they appear to be equal in this figure. I would suggest either adding scales, or at least clarifying that the graphical representations of contact area are not to scale.

Figure 2, superscript required for ‘cm2’ in the y-axis label.

Figure 3, the vertical alignment of all plots makes reading the results for the 500 and 600 kPa tests difficult. I would suggest staggering the results slightly left and right to make reading them easier to read.

Supplementary file, the Excel file could use a little formatting/clean-up to make it easier to interpret; for example, the headers for each section could be bolded/formatted as table headings; some headers are bolded already, but many aren’t

Reviewer #2: General comments:

This manuscript investigates the validity and day-to-day repeatability of an XSensor in-shoe pressure-sensing insoles during standardized loading protocols in comparison to known pressures and contact areas.

Overall, I agree with the authors on the importance for independent research groups to confirm the validity and reliability of commercial measurement systems. I also get the impression that the measurements were carried out carefully. In an original research article, however, the generated data should be presented and discussed in a way that provides value to other researchers. In my opinion, the submitted manuscript is lacking a clear structure, the investigated outcome variables are not well justified, and there are no criteria for what constitutes sufficient validity and reliability. My main critical points are:

1) The outcome variables are not well prepared and justified in the introduction. The authors talk about mean and peak pressure over varying duration, contact areas at different pressures, creep, insole consistency, etc.. It is unclear, why each variable is of interested and in which context each variable is relevant. The aim statement should clearly state the variables that the authors plan to use to determine the insoles’ validity and repeatability. Then, the variables should be described in the methods. Particularly, the authors should avoid to report results that were not prepared in the methods (creep, insole consistency).

2) What constitutes sufficient validity and repeatability? The authors talk about various applications of pressure-sensing insoles, e.g. running biomechanics [2] or in patients with neuromotor disorders [3-5]. Comparing pressures between slightly modified running shoes during running OR comparing pressures between healthy vs. diseased patients during standing or walking will lead to very different requirements of validity and repeatability for a pressure-sensing system. Therefore, the authors cannot make a sweeping statement in the abstract to say that “the Xsensor system is appropriate for clinical assessment that require multiple assessments”.

Therefore, the criteria that indicate sufficient validity and repeatability need to be developed carefully in the context of a certain research area and are only relevant in this context. Those criteria should not only be based on relative measures (e.g. relative error or ICC). Such measures certainly have their strengths but also weaknesses (e.g. Koo, Terry K., and Mae Y. Li. 2016. “A Guideline of Selecting and Reporting Intraclass Correlation Coefficients for Reliability Research.” Journal of Chiropractic Medicine 15 (2): 155–63.)

The authors should consider including absolute measures such as Bland-Altman limits of agreement (see Atkinson, G., and A. M. Nevill. 1998. “Statistical Methods for Assessing Measurement Error (reliability) in Variables Relevant to Sports Medicine.” Sports Medicine 26 (4): 217–38.). Limits of agreement would provide an intuitive way of determining whether the measurement error grows with increasing pressure or measurement duration.

Finally, the authors should come up with clinically relevant or meaningful changes in pressure in absolute physical units (or maybe relative to body weight) that occur in a certain research context and use those values to build a framework for validity and repeatability.

3) The authors criticize previous studies for their limited external validity. In this experiment however, there is no information on the loading rate. I assume that the pressure was built up slowly and was not comparable to a pressure profile to running or walking. Is there anyway for your system to simulate a dynamic loading profile? Currently, I would argue that the results of this manuscript are relevant for standing only.

Specific comments:

Line 72-73: ICCs of around 0.5 are not usually considered to show “high intra class correlation”.

Line 73-81: The relevance of previous findings on the Tekscan system is unclear for this study on the XSensor system. It would seem more relevant to explain if some measurement systems have a larger error than clinically relevant underfoot pressure changes in clinical populations.

Line 95: It is unclear how the “suitability for clinical and biomechanical assessment” is defined. There should be a-prior standards about the validity and repeatability for their use in clinical research and biomechanical research.

Line 99: When looking at the XSensor webpage (https://xsensor.com/), the available systems are “Intelligent Insoles Clinical” and “Intelligent Insoles Pro”. Please explain how the XSensor X4 system relates to these two available systems.

Line 115: How long were the measurements?

Line 117: Where exactly were those sensels and how many were excluded?

Line 118: sensel underlined?

Line 125: Please briefly describe the testing device to generate pressure during the regional protocol. How accurate is this device?

Line 129: How long were the measurements in the regional protocol?

Line 130: Where was the regional pressure applied?

Line 138: This is the first time that a between-day study design is mentioned. This must be mentioned in the aims of this study and at the beginning of the methods. In general, while it is appreciated that the previously developed protocol is cited [reference 18], the most important aspects of this protocol must still be mentioned. Like how many days passed between the measurements, etc.? Why were the durations 0,2, 10, 30 seconds selected? What is a duration of 0 seconds anyway? How exactly were relative and absolute errors calculated (e.g. the ones in Table 2)?

Lines 134-144: This section is difficult to follow. For example: What happened to the different pressures in the regional protocol (50, 110, 200 kPa)? Why were peak pressures investigated? Is “load” equal to pressure? Why did the authors only investigate ICCs as a relative measure of reliability? In my opinion, absolute measures of repeatability and validity such as Bland-Altman limits of agreement would be much more intuitive in their interpretation and it would be easier to observe changes in agreement with the known measures as a function of held duration or applied pressure. This section needs to be structured better and the analysis approach should be justified.

Line 143: Please explain exactly what type of ICC model was used and how the individual values were combined into the respective results. Different ICC models can lead to different results given the same data.

Figure 1: Please add a color scale so the reader can assess changes in pressure. Please indicate where in Figure 1a the regional pressures were applied.

Line 154: Why is there no analysis of the measured pressures in the regional protocol? This type of regional loading may affect the insole’s validity in measuring the correct mean and peak pressure.

Table 1: There is no description of this analysis in the methods.

Lines 160-167: It is unclear where those values come from? (e.g. S10 (23.09±15.92kPa; 9%)). Are those averages?

Line 164-165: There has been no mention of a time-dependent analysis of peak pressure in the methods section.

6. PLOS authors have the option to publish the peer review history of their article (what does this mean?). If published, this will include your full peer review and any attached files.

Reviewer #1: No

Reviewer #2: **Yes: **Maurice Mohr

---

## [Author Response · Author response to Decision Letter 0]

22 Jul 2022

We would like to thank the reviewers for their detailed and considered comments on our manuscript. We have made significant changes to the manuscript in response to comments which we believe has improved the quality and clarity of the data presented. Within this revision we have identified key changes which required reprocessing of the data. To prevent addition of similar and repeating figures or tables we have replaced prior versions with a more appropriate presentation of the data. We have also expanded the supplementary data which provides additional information to complement the results. 

We have two sections which respond to key points raised by the reviewers and are addressed below. We also have referred to these in the comments where they are relevant rather than repeating ideas and responses. 

Reliability response:

We appreciate the detailed review and critique from the reviewers and their constructive suggestion for improvements in our assessment of reliability. Following from this we have utilised a further assessment to assess reliability of peak and mean pressure values from the whole insole protocol, which allows us to assess these potential errors against a clinically relevant value (please see diabetes response below).

For the regional protocol we do not have a similar threshold of acceptable error as we do not have a known value where contact area error has clinical relevance. Further to this the data collected in this study does not allow assessment across multiple participants and so random error in this case is determined by a fixed count of sensors which overlap the indenter perimeter. This has led to us maintaining our approach with ICC for the assessment of reliability in the regional protocol, which has not been updated in the manuscript. The whole insole approach however is now described in detail in the methods with new comparisons of day 1 and day 2 presented in the results. 

As the purpose of this manuscript is to assess the suitability of the XSENSOR system for clinical measurement a clinically relevant analytical goal was deemed more suitable than a formal hypothesis test for the assessment of the measurement errors acceptability.

The approach described below and in the methods is taken from:

- G. Atkinson and A. M. Nevill, “Statistical methods for assessing measurement error (reliability) in variables relevant to sports medicine,” Sports Med., vol. 26, no. 4, pp. 217–238, 1998.

- A. M. Nevill, G. Atkinson, A. M. Nevill, and G. Atkinson, “Assessing agreement between measurements recorded on a ratio scale in sports medicine and sports science.,” Br. J. Sports Med., vol. 31, no. 4, pp. 314–318, Dec. 1997.

• Day one to day two was compared to each other and both were compared to the known pressure we applied through the TruBlue device. A percentage error was calculated.

• A T-test and correlation compared bias between day 1 and day 2 measurements.

• Bland-Altman plots with 95% agreement displayed disagreement between tests. 

• This data included heteroscedastic error and therefore natural logarithms were taken.

• Bias ratio was calculated based on the antilog of the mean difference between days.

• Limits of agreement were calculated as agreement ratios between which 95% of difference between days can be calculated 

• maximum reliability error (systematic bias and random error) was assessed against an analytical goal of 15% 

This assessment of the data has enabled us to describe in the results and discussion the limits of agreement with reference to the MID and an analytical goal. Therefore, with relevance to the clinical assessment of patients with diabetes (please see response below). 

Clinical assessment in Diabetes response: 

We have narrowed down the aims of the research study to focus on diabetes and therefore have contextualised the reliability and validity of pressures systems with relevance to this application of the technology: 

There are two IWGDF guidelines which refer to expectations of pressure reduction.

- 30% reduction is threshold for effectiveness in offloading footwear and interventions

- 200 kPa is a threshold for risk of recurrent ulceration 

We have incorporated these into our paper introduction and described our method for application and implementation of these in the method. Using the 30% reduction guideline we have defined an analytical goal of 15% reliability error (systematic bias and random error) such that each days measured value should not be affected by more than half of the objective reduction.

Specific responses to reviewer comments (also provided in response to reviewers document):

Reviewer #1 

R1C1. The authors’ approach to quantifying accuracy is well founded and executed. However, I think the approach to day-to-day reliability is not as rigorous. I am generally skeptical of ICCs and their use in quantifying reliability as they have many issues that have been identified in literature (Muller and Buttner, 1994). My suggestion would be to reframe the problem of reliability as a hypothesis test with respect to a minimally important difference (MID).

We thank the reviewer for their thorough and constructive reviews. Please see responses below that reference these points in more detail. In short: we have adopted this approach of defining a clinically relevant MID and testing the outcomes from our protocol against this. 

R1C2. The authors have provided a reference that suggests clinically important changes in pressure would be on the order of 30%. Using this (or other prior work as appropriate), we can define an MID that we would expect to see in a clinical study. If we wish to be conservative, we could use the lower end of the typical range of pressure values, 100 kPa, and the reference of 30% gives us an MID of 30 kPa. We can then take the measurements for day 1 and day 2, across all pressure conditions, and treat these as repeated measures observations. In doing so, we can calculate the distribution of the differences between days and conduct a single-sided hypothesis test with respect to an MID of 30 kPa (the alternate hypothesis being that our estimated mean of differences exceeds 30 kPa).

The advantage of this approach is that it provides a concrete benchmark for evaluating whether the insoles will measure pressure reliably. In my opinion, this is more informative than ICCs, and a more statistically rigorous test of insole reliability. I also suspect that the findings would favour the insoles as being highly reliable. 

An analytical goal approach was employed in place of a formal hypothesis test.

Please see the response to key points with more detail about the reliability assessment and the clinical relevance. 

R1C3. A table that provides comparative values for the XSENSOR system, similar to Table 1 from Price et al., 2016, might be of interest to the reader.

We provide the details of the XSENSOR system within the first line of the method and haven’t included a table as there is nothing to compare this to within this paper. 

We have expanded this content to include more technical information to make the information provided comparable to that in the prior paper. 

R1C4. Introduction

There are somewhat contradictory statements as well. It is stated that comparing between different studies/systems is very difficult, while the aim of the current study is to provide such a comparison between XSENSOR and other systems. Yet, this study does not address this difficulty at all; it only provides similar quantifications for the XSENSOR as has already been done for other systems in earlier work. I’d suggest a revision of the introduction to make it more focused on the purpose of the article: evaluating the XSENSOR system for clinical use.

The content of the introduction and aims have been amended and altered as suggested. This is now to focus on clinical assessments in diabetes and appropriate standards of validity and repeatability to enable this. The detail of the introduction has been increased to more specifically relate to variables of interest in a clinical context and then, more specifically, in Diabetes. We have continued to include a range of variables (such as creep, consistency) in the introduction to provide context in terms of the work already done to validate pressure systems. The variables utilised within this work are identified more clearly as clinically relevant and then fully defined and described in the methods.

R1C5. Throughout the manuscript there are no spaces between numbers and units and these should be added throughout (e.g., 400kPa should be 400 kPa).

The manuscript has been amended as suggested

R1C6. Line 53: capitalization of diabetic not necessary

The manuscript has been amended as suggested

R1C7. Line 60: single quotes around creep not required; this is a standard physics term

The manuscript has been amended as suggested

R1C8. Line 106: single quotes around calibration file not required

The manuscript has been amended as suggested

R1C9.Line 108: single quotes around active sensels not required

The manuscript has been amended as suggested

R1C10. Line 126: this sentence is a little unclear as it may read as though there were two separate cylinders with different bases; perhaps consider rewording 

Wording has been changed to:

“Two cylinders were used to apply regional loads in this study. Each cylinder was made of rigid plastic and had a 1mm layer of PORON at its base.”

R1C11. Line 130: “For [each of] the six conditions…”

Amended as suggested

R1C12. Line 138 and 144: As mentioned in my general comments, ICCs are not the ideal representation of reliability

Please see full reliability response above. 

R1C13. Line 144: suggest “performed” instead of “undertaken”

Amended as suggested

R1C14. Line numbers stopped after page 10 

We apologise for this error; line numbering has been corrected in the revised submission. 

R1C15. Page 11, First paragraph, the peak pressures for 100-400 kPa measurements seem to be within 1-8 % of the target measurements, unless there is some averaging done between day 1 and 2.

This has been updated to reflect the range across both days.

“insoles recorded peak pressure measurement errors of 1-8%.”

R1C16. Page 11, First paragraph, “… interact with evaluations of efficacy…” might be clearer as “… bias evaluations of efficacy…”

The manuscript has been amended as suggested

R1C16. Page 11, First paragraph, while some of the mean measurements did indeed show less error than peaks, this was not universally true; I suggest rephrasing this statement

This paragraph has been reworded to more clearly represent the full range presented in figure 2 when observed across both days.

R1C17. Page 13, Second paragraph, given that the testing in this study was done under very controlled laboratory conditions, the evaluation of reliability should reflect the caveat that calibration and measurement under in situ conditions might be less reliable than what has been shown

Detail added to paragraph 

“These data however, have the limitation that they were collected within controlled laboratory conditions and the reliability of the system for clinical use should be explored in-situ with relevant patients also.”

R1C18. Figure 1, the scales for the grids in this figure is not obvious, and if I am correct, the scales are not identical between S4 and S10, even though they appear to be equal in this figure. I would suggest either adding scales, or at least clarifying that the graphical representations of contact area are not to scale.

We have now revised figure 1 to include a clearer representation of the data. 

Size 10 insoles each pixel represents 0.82cm x 0.82cm while Size 4 insoles each pixel size represents 0.7cm x 0.7cm. 

To allow direct comparison the images have now been scaled to reflect the true dimensions. To further aid comparison to the target indenter area a scale outline of the indenter has been added centred on the middle of the active area. 

R1C19. Figure 2, superscript required for ‘cm2’ in the y-axis label.

The figure has been amended as suggested

R1C20. Figure 3, the vertical alignment of all plots makes reading the results for the 500 and 600 kPa tests difficult. I would suggest staggering the results slightly left and right to make reading them easier to read.

Figure has been amended. Data has been staggered slightly such that it is offset for each pressure increment within each time of application to prevent the overlap of standard deviation bars and peak values. 

R1C21. Supplementary file, the Excel file could use a little formatting/clean-up to make it easier to interpret; for example, the headers for each section could be bolded/formatted as table headings; some headers are bolded already, but many aren’t

Formatting has been improved throughout the document. 

Reviewer #2: General comments:

R2C1. The outcome variables are not well prepared and justified in the introduction. The authors talk about mean and peak pressure over varying duration, contact areas at different pressures, creep, insole consistency, etc.. It is unclear, why each variable is of interested and in which context each variable is relevant. The aim statement should clearly state the variables that the authors plan to use to determine the insoles’ validity and repeatability. Then, the variables should be described in the methods. Particularly, the authors should avoid to report results that were not prepared in the methods (creep, insole consistency).

We have continued to include a range of variables (such as creep, consistency) in the introduction to provide context in terms of the work already done. The detail of the introduction has been increased to more specifically relate to variables that are of interest in a clinical context and then, more specifically, in Diabetes. 

The plantar pressure variables utilised within this work are identified more clearly as clinically relevant and then fully defined and described in the methods, aligning each outcome measure with the purpose for it’s conclusion. 

R2C2. What constitutes sufficient validity and repeatability? The authors talk about various applications of pressure-sensing insoles, e.g. running biomechanics [2] or in patients with neuromotor disorders [3-5]. Comparing pressures between slightly modified running shoes during running OR comparing pressures between healthy vs. diseased patients during standing or walking will lead to very different requirements of validity and repeatability for a pressure-sensing system. Therefore, the authors cannot make a sweeping statement in the abstract to say that “the Xsensor system is appropriate for clinical assessment that require multiple assessments”.

We have slightly reworded this sentence at the end of the abstract to say it is ‘likely’. The sentence was specific to the repeatability between day as described by the clinical assessments that require multiple assessments. We hope that the addition of the more detail exploration of reliability as recommended by the reviewers also further supports this statement. 

R2C3. Therefore, the criteria that indicate sufficient validity and repeatability need to be developed carefully in the context of a certain research area and are only relevant in this context. Those criteria should not only be based on relative measures (e.g. relative error or ICC). Such measures certainly have their strengths but also weaknesses (e.g. Koo, Terry K., and Mae Y. Li. 2016. “A Guideline of Selecting and Reporting Intraclass Correlation Coefficients for Reliability Research.” Journal of Chiropractic Medicine 15 (2): 155–63.)

The authors should consider including absolute measures such as Bland-Altman limits of agreement (see Atkinson, G., and A. M. Nevill. 1998. “Statistical Methods for Assessing Measurement Error (reliability) in Variables Relevant to Sports Medicine.” Sports Medicine 26 (4): 217–38.). Limits of agreement would provide an intuitive way of determining whether the measurement error grows with increasing pressure or measurement duration.

Please see full reliability response.

R2C4. Finally, the authors should come up with clinically relevant or meaningful changes in pressure in absolute physical units (or maybe relative to body weight) that occur in a certain research context and use those values to build a framework for validity and repeatability.

As detailed above, we have added further detail from prior research relating to absolute measures that are relevant. We have also increased the specificity of the research question to relate to clinical assessment in diabetes to ensure that the framework for reliability, validity is clear and focussed on MID. 

R2C5. The authors criticize previous studies for their limited external validity. In this experiment however, there is no information on the loading rate. I assume that the pressure was built up slowly and was not comparable to a pressure profile to running or walking. Is there anyway for your system to simulate a dynamic loading profile? Currently, I would argue that the results of this manuscript are relevant for standing only.

Loading was applied as quickly as possible but was done by hand (using a pressure release valve) so was not at a controlled rate. Detail has been added to the method to clarify this: 

“Each pressure was applied manually as quickly as possible using a hand release valve to inflate a bladder.”

This was not at the same rate as unloading and loading the foot would be in stance for walking or running. But took approximately 1-2 seconds to be applied. From our prior studies (Price 2016) we would not expect drift in pressure systems over time to affect the measured pressure.

Using this device, it is not possible to simulate a dynamic loading profile. As we are applying load to the target rapidly over a short period of time (typically 2s) we believe that the instant of target pressure and post 2s of held load have relevance to gait

While we agree longer duration loading (10s and over) are relevant to standing the instant of and post 2s measures can be associated with gait in healthy and pathological cohorts. The 2s cohort has been selected for the main analysis as this is unlikely to be affected by any instant reaction to loading or by system drift over the short duration. 

Collecting data over a longer duration also enables researchers to be aware of how protocols with more prolonged loading may influence the reliability and validity. This is relevant as more real world data collection occurs and the influence of other tasks (such as standing) potentially becomes a relevant aspect of potential intervention effectiveness tasks in the future. 

R2C6. Line 72-73: ICCs of around 0.5 are not usually considered to show “high intra class correlation”. 

Removed as per removal of ICC. 

R2C7. Line 73-81: The relevance of previous findings on the Tekscan system is unclear for this study on the XSensor system. It would seem more relevant to explain if some measurement systems have a larger error than clinically relevant underfoot pressure changes in clinical populations.

We have added further detail relating to the requirements of systems for clinical assessment to the introduction. This includes values and numbers that offer context to these numbers for direct comparison. 

R2C8. Line 95: It is unclear how the “suitability for clinical and biomechanical assessment” is defined. There should be a-prior standards about the validity and repeatability for their use in clinical research and biomechanical research.

This section has been rewritten with an analytical goal based on a relevant meaningful difference. This is now defined within the methods and presented within the results as per the suggestion of reviewer one. 

R2C9. Line 99: When looking at the XSensor webpage (https://xsensor.com/), the available systems are “Intelligent Insoles Clinical” and “Intelligent Insoles Pro”. Please explain how the XSensor X4 system relates to these two available systems.

The XSENSOR X4 is the insole sensor version. 

Both the clinical and pro systems supplied by XSENSOR use the same insole sensors used within this study. The difference between systems is the on-shoe hub device which has increased memory capacity in the pro version. 

R2C10. Line 115: How long were the measurements?

Further detail has been added to the regional protocol section to define the measurements as 30 seconds long. 

R2C11. Line 117: Where exactly were those sensels and how many were excluded?

Sentence reworded to include location as:

Based on manufacturer guidance some sensels on the insole periphery were excluded due to raised cabling enclosure which affected the even application of pressure.

R2C12. Line 118: sensel underlined? 

Underline removed 

R2C13. Line 125: Please briefly describe the testing device to generate pressure during the regional protocol. How accurate is this device?

The following details have been added

“Two cylinders were used to apply regional loads in this study. Each cylinder was made of rigid plastic and had a 1mm layer of PORON at its base.”

“Cylinders were loaded by the application weights to their top surface and supported by a vertical linear bearing to ensure load through their centre to generate realistic pressures for these anatomical regions during stance of 50, 110 and 200 kPa.”

Pressures applied were based on loads divided by contact area for each indenter. Mass applied by the cylinder was validated using a force plate. Measurement of absolute pressure was not the intended outcome of this assessment and the three load levels were chosen to represent low moderate and high load for each area. 

R2C14. Line 129: How long were the measurements in the regional protocol?

The duration of recording for the regional protocol has been added to the method.

 “three measurements were recorded of 30 seconds,”

R2C15. Line 130: Where was the regional pressure applied?

Detail has been added to describe the location of the device in the rearfoot, where we would anticipate the heel to be. 

This has also been included in Figure 1.

R2C16. Line 138: This is the first time that a between-day study design is mentioned. This must be mentioned in the aims of this study and at the beginning of the methods. In general, while it is appreciated that the previously developed protocol is cited [reference 18], the most important aspects of this protocol must still be mentioned. Like how many days passed between the measurements, etc.? Why were the durations 0,2, 10, 30 seconds selected? What is a duration of 0 seconds anyway? How exactly were relative and absolute errors calculated (e.g. the ones in Table 2)?

The introduction now includes further detail relating to within- and between day in relevant sections including the aim.

The methods section has been made more detailed and includes further detail of the original protocol. 

The testing was undertaken on consecutive days. 

The duration of 0 seconds was a measure that was taken as soon as the pressure insole was loaded to the correct pressure magnitude for each measure. As described in response to the other reviewer comment, this was around 1- 2 seconds. This has now been described in the regional protocol section: 

“The start point for measurement was when the target pressure was achieved (monitored by a pressure gauge “ 

and the variables section of the method. 

“T0 reflected a measure at the first instant at which the required pressure had been applied and the other T values added to this.”

Within this manuscript to align to the focus of use for clinical interpretation of risk we have used the 2s time point as discussed in response to the comment above. 

The longer durations (T10 & T30) were selected to reflect changes with the development in technology in the field, potable assessment of pressure is more viable and more common. Therefore, we undertook a data collection which facilitated generalising results to longer stance times than a walking gait, such as standing tasks which are also relevant to loading the at-risk foot. We refer to this literature in the discussion of the paper where we also address higher pressures for example: 

This is particularly relevant with the trend in more real-world data collection which is arising in more recent literature, where peak pressures may exceed those recorded in walking 

The calculation of the specific errors has been added to the method, variables section for clarity:

“Peak and mean pressure error were calculated based on the absolute value versus the target pressure, error was also presented as a percentage of the target pressure”.

R2C17. Lines 134-144: This section is difficult to follow. For example: What happened to the different pressures in the regional protocol (50, 110, 200 kPa)? Why were peak pressures investigated? Is “load” equal to pressure? Why did the authors only investigate ICCs as a relative measure of reliability? In my opinion, absolute measures of repeatability and validity such as Bland-Altman limits of agreement would be much more intuitive in their interpretation and it would be easier to observe changes in agreement with the known measures as a function of held duration or applied pressure. This section needs to be structured better and the analysis approach should be justified.

The methods section has been made more detailed and includes further detail of the original protocol. This details the assessment of mean and peak pressures. 

Re ICC: Please see full reliability response.

R2C18. Line 143: Please explain exactly what type of ICC model was used and how the individual values were combined into the respective results. Different ICC models can lead to different results given the same data.

This text has been removed alongside other amendments and removal of ICC. Please see full reliability response. 

R2C19. Figure 1: Please add a colour scale so the reader can assess changes in pressure. Please indicate where in Figure 1a the regional pressures were applied. 

Figure 1 has been updated

For Figure 1 (a) has been updated to include an outline of the indenter location on each insole. 

For Figure 1 (b) this shows the difference from the target pressure in % (which lies in the range of +/-15%) 

For Figure 1 (c) a colour bar has been added showing the variation in load across the indenter area this is represented in % with full scale of +/-100%. As the number of active sensors and cumulative area is the outcome of interest in this image a to scale outline of the indenter has been added centred on the middle of the active area. 

R2C20. Line 154: Why is there no analysis of the measured pressures in the regional protocol? This type of regional loading may affect the insole’s validity in measuring the correct mean and peak pressure. 

The regional protocol is intended to assess loaded contact area under different applied pressures. 

As described above pressures were produced by application of a mass to the cylinder to represent a low moderate and high load over the cylinder’s contact area. 

The application of different pressures allows us to assess whether the activation or sensitivity of sensors overlapping the border of the cylinder is dependent on the applied load. Therefore whether the contact area measures of the insoles are reliant on the magnitude of the pressure being applied over the areas. 

While in theory It would be possible to calculate a total force applied across the contact area this is not the purpose of this test. 

R2C21. Table 1: There is no description of this analysis in the methods.

The analysis of the data to derive this data has been added to the method, in the variables section: 

“finally, the percentage of sensors within 5% of the insole mean was calculated to allow assessment of variation across the test insole.”

R2C22. Lines 160-167: It is unclear where those values come from? (e.g. S10 (23.09±15.92kPa; 9%)). Are those averages?

We have defined these variables more precisely in the method, as previously described. Also we have improved he description in the sentence and added a reference to a figure and a table which both include the relevant data: 

“Peak pressure was higher than the target in S4 (Mean ± SD and Error from target across all pressure levels: 20.64 ± 13.69 kPa; 7%) and S10 (23.09 ± 15.92 kPa; 9%) with errors from the target highest at 50 kPa (13% and 20% respectively) (Figure 3; Table 1).”

R2C23. Line 164-165: There has been no mention of a time-dependent analysis of peak pressure in the methods section. 

The method and results have been restructured and further detail added with clear definition of main outcome variables and presentation of results.

---

## [Decision Letter · Decision Letter 1]

14 Sep 2022

PONE-D-21-30139R1Validity and reliability of the XSENSOR in-shoe pressure measurement systemPLOS ONE

Dear Dr. Parker,

Thank you for submitting your manuscript to PLOS ONE. We invite you to submit a revised version of the manuscript that addresses the remainig points raised during the review process. Both reviewers point out a number of small issues. Please adress them carefully.

We look forward to receiving your revised manuscript.

Kind regards,

Peter Andreas Federolf

Academic Editor

PLOS ONE

Journal Requirements:

Reviewers' comments:

Reviewer's Responses to Questions

**Comments to the Author**

1. If the authors have adequately addressed your comments raised in a previous round of review and you feel that this manuscript is now acceptable for publication, you may indicate that here to bypass the “Comments to the Author” section, enter your conflict of interest statement in the “Confidential to Editor” section, and submit your "Accept" recommendation.

Reviewer #1: (No Response)

Reviewer #2: (No Response)

2. Is the manuscript technically sound, and do the data support the conclusions?

Reviewer #1: Yes

Reviewer #2: Yes

3. Has the statistical analysis been performed appropriately and rigorously? 

Reviewer #1: Yes

Reviewer #2: Yes

4. Have the authors made all data underlying the findings in their manuscript fully available?

Reviewer #1: Yes

Reviewer #2: Yes

5. Is the manuscript presented in an intelligible fashion and written in standard English?

Reviewer #1: Yes

Reviewer #2: Yes

6. Review Comments to the Author

Reviewer #1: Thank you to the authors for their comprehensive response to the comments. I agree that the manuscript is much improved and I appreciate them taking the time to revise the content and analysis.

I have only a few minor remaining comments based on the revised version with changes highlighted

Line 76: “discrete” misspelled

Line 97: “that” seems superfluous

Line 98: wording is a bit unclear; is it supposed to be “highly correlated relationships”?

Line 108: capitalization of international not needed

Line 120: missing a period.

Lines 142-143: inclusion of pressure gauge company redundant with information in previous paragraph.

Line 145: change to “were calculated”

Lines 224-225: I am confused by the statement that no measurements exceeded the analytical goal, when one of the 50 kPa measurements was off by 20%.

Line 340: change to “these data”

Reference 11: misspelled “pressure”

Reviewer #2: General comments:

I would like to congratulate the authors on the extensive revisions to the manuscript. The inclusion of Bland-Altman limits of agreement and a clinically meaningful reliability target has significantly increased the value of this manuscript to the readership. Please consider a few remaining but minor comments/questions below. ^

Specific comments:

Line 69: discrete

Line 74: average pressure across sensors within a discrete region

Line 96-102: I am still under the impression that this part of the introduction does not add value to the manuscript. The importance of evaluating the contact area has been demonstrated further up already. I do not understand the meaning of “Additionally, that when considering repeated measures most systems demonstrate high relationships between the two days or measures”.

Line 173: intra-class correlation coefficient

Line 220-221: This sentence sounds like a contradiction. 20% > 15%

Figure 1: Is there any reason for the measurement at 100kPA showing the opposite trend of all other loading conditions? On average Figure 3 does not seem to show this opposite behavior at 100kPA.

Figure 4: (c) and (d) labels are flipped in the figure

Table 1&2: Does the correlation column represent the correlations before or after log-transformation? After transformation, the correlations should be much lower or rather non-existent. Also, there are strong negative and positive correlations (e.g. Table 1, s10, 100kPA vs. 500kPA). Do the authors have an explanation for this? I would expect a consistent trend where d2 shows higher pressures than d1 across all loading conditions?

7. PLOS authors have the option to publish the peer review history of their article (what does this mean?). If published, this will include your full peer review and any attached files.

Reviewer #1: No

Reviewer #2: **Yes: **Maurice Mohr

---

## [Author Response · Author response to Decision Letter 1]

25 Oct 2022

We would like to thank the reviewers for their careful review of this manuscript.

please see response to reviewer document attached to this submission.

---

## [Editor Report · Decision Letter 2]

8 Nov 2022

Validity and reliability of the XSENSOR in-shoe pressure measurement system

PONE-D-21-30139R2

Dear Dr. Parker,

We’re pleased to inform you that your manuscript has been judged scientifically suitable for publication and will be formally accepted for publication once it meets all outstanding technical requirements.

Kind regards,

Peter Andreas Federolf

Academic Editor

PLOS ONE
---

## [Editor Report · Acceptance letter]

25 Nov 2022

PONE-D-21-30139R2 

Validity and reliability of the XSENSOR in-shoe pressure measurement system 

Dear Dr. Parker:

I'm pleased to inform you that your manuscript has been deemed suitable for publication in PLOS ONE. Congratulations! Your manuscript is now with our production department. 

Kind regards, 

on behalf of

Dr. Peter Andreas Federolf 

Academic Editor

PLOS ONE